# Quantitative Measurement of Functional Activity of the PI3K Signaling Pathway in Cancer

**DOI:** 10.3390/cancers11030293

**Published:** 2019-03-01

**Authors:** Anja van de Stolpe

**Affiliations:** Precision Diagnostics, Philips Research, High Tech Campus, 5656AE Eindhoven, The Netherlands; anja.van.de.stolpe@philips.com

**Keywords:** signal transduction pathway, PI3K, FOXO, assay, Bayesian model, mRNA, target gene, oxidative stress, crosstalk, cancer, immune response

## Abstract

The phosphoinositide 3-kinase (PI3K) growth factor signaling pathway plays an important role in embryonic development and in many physiological processes, for example the generation of an immune response. The pathway is frequently activated in cancer, driving cell division and influencing the activity of other signaling pathways, such as the MAPK, JAK-STAT and TGFβ pathways, to enhance tumor growth, metastasis, and therapy resistance. Drugs that inhibit the pathway at various locations, e.g., receptor tyrosine kinase (RTK), PI3K, AKT and mTOR inhibitors, are clinically available. To predict drug response versus resistance, tests that measure PI3K pathway activity in a patient sample, preferably in combination with measuring the activity of other signaling pathways to identify potential resistance pathways, are needed. However, tests for signaling pathway activity are lacking, hampering optimal clinical application of these drugs. We recently reported the development and biological validation of a test that provides a quantitative PI3K pathway activity score for individual cell and tissue samples across cancer types, based on measuring Forkhead Box O (FOXO) transcription factor target gene mRNA levels in combination with a Bayesian computational interpretation model. A similar approach has been used to develop tests for other signaling pathways (e.g., estrogen and androgen receptor, Hedgehog, TGFβ, Wnt and NFκB pathways). The potential utility of the test is discussed, e.g., to predict response and resistance to targeted drugs, immunotherapy, radiation and chemotherapy, as well as (pre-) clinical research and drug development.

## 1. Introduction

A limited number of signal transduction pathways are evolutionarily well conserved to control basic cellular processes, such as cell division, cell differentiation and migration [1,2,3]. Signaling pathways can be roughly categorized as hormone driven nuclear receptor pathways (e.g., androgen and estrogen receptor pathways), developmental pathways (e.g., Wnt, Hedgehog, TGFβ, and Notch pathways), the inflammatory NFκB pathway, and the highly complex growth factor regulated signaling pathway network, in which the phosphoinositide 3-kinase (PI3K)-AKT-mTOR pathway is probably the most prominent, next to MAPK and JAK-STAT pathways [2,4,5,6,7,8,9,10,11]. These signaling pathways are core regulators of embryonic development and control important physiological processes, such as hematopoiesis and generation of an immune response, tissue regeneration, hair growth, and renewal of intestinal mucosa [11,12,13,14]. Naturally, in physiological cellular processes, signaling pathway activity, as well as crosstalk between different pathways, is tightly controlled [15,16,17,18]. However, they also play important roles in the pathophysiology of many, if not all, diseases, for example, benign and malignant tumors, auto-immune (e.g., rheumatoid arthritis) and immunodeficiency diseases, and neurological diseases (e.g., epilepsy) [7,9,11,19,20,21,22,23,24,25,26,27,28]. In diseases, control of their activity is typically altered or lost. In cancer, this is exemplified by uncontrolled cell division compared to a strictly controlled balance between cell division and differentiation in the developing embryo, while distant metastasis can be seen as an uncontrolled version of cell migration during organ development [1,29]. These cancer characteristics constitute important hallmarks of cancer [22].

## 2. The PI3K Pathway in Oncology

From a clinical perspective, the PI3K-AKT-mTOR signal transduction pathway is certainly one of the most important signaling pathways as it regulates key processes such as cell division, migration, metabolism, and DNA repair in a large variety of cell types, to an important extent also by recruiting and influencing the activity of one or more of the other signaling pathways [30,31,32,33]. In cancer, the PI3K pathway is considered a proliferation and survival pathway [22]. In addition, activation of the PI3K-AKT pathway has been associated with resistance to multiple therapeutic modalities, e.g., radiation and chemotherapy, and hormonal therapy [34,35,36,37].

The PI3K pathway consists of a membrane receptor, which, upon binding an appropriate ligand, activates a downstream PI3K signaling molecule, resulting in the activation of the AKT and mTOR effector proteins and the AKT-mediated inhibition of the Forkhead Box O (FOXO) transcription factor. Membrane receptors are G-protein coupled receptors (GPCR) such as the β_2_-adrenergic receptor with its ligand epinephrine, and members of the receptor tyrosine kinase family (RTK, e.g., human epidermal growth factor receptors HER2 and HER3) with ligands such as the epidermal growth factor (EGF) and heregulin. Multiple isoforms of PI3K signaling enzymes exist. Class I PI3K isoforms are relevant for cancer, i.e., p110α (encoded by the *PIK3CA* gene), p110β (*PIK3CB* gene), p110γ (*PIK3CG* gene), and p110δ (*PIK3CD* gene). They catalyze the formation of the phospholipid phosphatidylinositol (3,4,5) triphosphate (PtdIns(3,4,5)*P*3 or PIP3) by the addition of a third phosphate group to the phosphatidylinositol bisphosphate (PtdIns-4,5-*P*2 or PIP2) [30]. Phosphatases such as PTEN exert an inhibitory effect on the pathway by removing the third phosphate group from PIP3. PIP3 phospholipids are generated close to the cell membrane where they attract proteins such as AKT that contain a specific PIP3-binding group and become activated. Which PI3K isoform(s) is (are) used by a malignant cell depends on the cell type of origin and the presence of PI3K-activating genomic mutations. For example, immune cells dominantly use p110γ and p110δ isoforms, while genomic loss of the tumor suppressor protein PTEN is thought to result in specific activation of the p110β isoform [6,30]. Complex negative feedback mechanisms function within the PI3K pathway [30].

The PI3K pathway is linked via the activation of AKT kinase to FOXO transcription factors. FOXO family members are ubiquitously expressed transcription factors [31,33,38,39,40]. They transcribe their target genes by recognizing and binding to two different promoter enhancer sites, the Daf-16 family member-binding element (DBE, 5′-GTAAA(T/C)AA-3′) and the insulin-responsive element (IRE, 5′-(C/A)(A/C)AAA(C/T)AA-3′) [41]. In healthy non-dividing tissue, FOXO’s are located in the nucleus and when active they transcribe target genes that are necessary to maintain a non-dividing differentiated cellular phenotype, e.g., multiple cell cycle kinase inhibitors [33]. Upon activation of the PI3K pathway, phosphorylated AKT inhibits the transcriptional activity of the FOXO transcription factor, causing translocation from the nucleus to the cytoplasm, and thereby enabling cell division.

Phosphorylated AKT also mediates the activation of the mammalian target of rapamycin complex 1 (mTORC1), which facilitates the assembly of the eIF4F protein complex required for protein translation [30,42]. This results in increased protein translation from a subset of mRNAs, including survival (e.g., *BCL-2* family) and proliferation (e.g., cyclins) proteins [43]. By this mechanism, activation of the PI3K pathway can lead to the amplification of pro-tumorigenic effects of co-active signaling pathways that transcribe target genes, such as the above mentioned signal transduction pathways.

In addition, activation of the PI3K pathway can lead directly or indirectly to activation of other signaling pathways, for example growth factor pathways, such as the ERK-MAPK and JAK-STAT pathways, but also developmental pathways, for example, Wnt, TGFβ, and Hedgehog pathways [16,30,44,45,46,47,48]. As an example, the TGFβ pathway exerts an anti-proliferative effect in the absence of an active PI3K pathway, mediated by the transcriptional activity of the TGFβ pathway-associated SMAD transcription factor together with FOXO. Upon activation of the PI3K pathway, this tumor suppressive effect of TGFβ can be lost due to the cytoplasmic translocation of FOXO, or even switched towards a tumor-promoting effect in the presence of an active MAPK-AP1 pathway [45,46,47]. Crosstalk between the PI3K pathway and the MAPK and JAK-STAT signaling pathways is common and can take place through (direct) interaction between expressed signaling molecules of the different pathways. As a consequence, these growth factor pathways are often not active in an isolated manner [30,49,50,51].

The PI3K pathway is the most frequently activated signaling pathway in cancer. In physiological processes in healthy cells the pathway is typically activated by growth factors. However, in many types of cancer the pathway is activated by genomic aberrations, such as gene amplification (e.g., *HER2*), gene deletion or loss-of-function mutations in genes encoding inhibitory proteins (e.g., *PTEN*), or specific activating mutations in one of the key pathway-activating components (e.g., *EGFR*, *HER2*, *PIK3CA*) [52]. In many experimental cancer model systems, like cell line culture or xenograft mice, growth factors may be abundantly present to activate the PI3K pathway, also in the absence of activating genomic changes on the premise that the necessary pathway proteins are expressed [53,54]. 

Of high relevance to cancer in view of recent developments in immunotherapy is the role of the PI3K pathway in the differentiation of immune cells and the generation of a proper anti-cancer immune response [30,55]. Immune cells predominantly use p110γ and p110δ isoforms to regulate complex immune-specific functions, often involving crosstalk with other signaling pathways, such as JAK-STAT and NFκB pathways. The PI3K pathway controls important and cell type-specific functions in cells of both the innate and adaptive immune system, e.g., proliferation, subtype differentiation, migration, generation of reactive oxygen species, and lymphocyte activation via B and T-cell receptors [30].

## 3. Drugs to Modify the Activity of the PI3K Pathway

Targeted drugs that inhibit the PI3K pathway are clinically available and more are in development stages or in clinical trials for many types of cancer [56,57,58]. They aim to inhibit the pathway at specific locations, depending on the underlying pathophysiology. Drug targets are the membrane receptor (e.g., trastuzumab), one or more of the PI3K isoform enzymes (targeted by "panPI3K" inhibitors and isoform-selective inhibitors), AKT, or mTOR (rapamycin analogs), or combinations of these signaling molecules (e.g., dual PI3K-mTOR inhibitor).

In cancer, drugs targeting the PI3K pathway are generally only effective in a subgroup of patients, and their use is associated with frequent resistance and more or less severe side effects [30,59]. Resistance can be either primary, being present at the start of the therapy, or develop during PI3K pathway inhibition therapy. Primary resistance can be due, for example, to the PI3K pathway being inactive, a pathway-activating mutation being present downstream of the drug target, or FOXO-induced protection against apoptosis; acquired resistance can arise from the inhibition of negative feedback loops, switching of PI3K isoforms, or activation of crosstalk with other pathways that take over the cell division function [30,42,56,57,59]. Neither type of resistance can currently be predicted reliably. In general, side effects can be inferred from the known role of the PI3K pathway in the specific cell or tissue type, and they tend to increase with use of less specific drugs (e.g., a pan-PI3K inhibitor) and dual target drugs (e.g., an mTOR-PI3K inhibitor) [60]. For example, many side effects stem from disrupting the delicate immune balance and interference with normal glucose metabolism, while neuropsychiatric side effects are relatively common for drugs that cross the blood brain barrier [60]. 

## 4. Measuring Functional PI3K Pathway Activity

Tests to measure PI3K pathway activity in cancer tissue or cell samples are needed to improve prediction (and monitoring) of PI3K pathway-targeted therapy response in patients [56,61]. Human epidermal growth factor receptor 2 (HER2) immunohistochemistry (IHC) and *HER2* fluorescent in situ hybridization (FISH) tests are available to select patients for HER2-inhibitory drugs such as trastuzumab. When correctly performed and interpreted they are good predictors for response to these drugs [62]. However, in breast cancer, for example, the test is inconclusive in at least 20% of patients, and a decision with respect to targeted therapy cannot be taken [62]. Moreover, HER2 testing cannot be used to predict response to other PI3K pathway inhibitors [57]. Gene mutations (e.g., *PIK3CA*, *AKT* mutations, *PTEN* loss) and AKT phosphorylation status appeared not to be sufficiently precise in predicting response to PI3K pathway inhibitors, while to the best of our knowledge predictive mRNA profiles are not clinically available [62,63,64].

We have recently described a novel approach to develop tests to quantitatively measure the functional activity of signaling pathways in individual tissue and cell samples, across cancer types [65,66]. Using this approach we recently reported the development and biological validation of a test for PI3K pathway activity, based on measuring the activity of the FOXO transcription factor as an inverse readout of PI3K pathway activity [39,67,68]. In brief, FOXO activity is inferred from mRNA expression levels of 26 high evidence FOXO target genes using a knowledge-based Bayesian network computational model (Figure 1). mRNA levels that serve as input for the computational model were obtained from Affymetrix expression microarray data. This initial PI3K pathway test was successfully biologically validated on multiple cell and tissue types for which the state of FOXO activity was known and derives a highly quantitative PI3K pathway activity score from the calculated FOXO activity score [67] (example, Figure 2). It has since been adapted from Affymetrix expression microarray (HG-U133-Plus2.0) to qPCR measurements of the target gene mRNA levels to enable use on formalin-fixed paraffin-embedded tissue samples, being the routinely available material in pathology diagnostics. Translation to use with RNA sequencing data is in progress. In general in non-dividing healthy tissue, FOXO is measured as active and the PI3K pathway is measured as inactive. FOXO is frequently inactive in dividing cells associated with an active PI3K pathway, for example in benign or malignant tumors or in clonally expanding lymphocytes, and becomes active when cells are treated with drugs that inhibit the PI3K pathway [64,67]. 

Although activity of the FOXO transcription factor can be reliably measured with this method, interpretation with respect to PI3K pathway activity is not always straightforward. In cancer tissue, FOXO can be alternatively activated by cellular oxidative stress [39]. Oxidative stress can be caused by an active PI3K pathway (potentially in combination with other active growth factor pathways) causing rapid cell division and metabolic changes, resulting in high levels of reactive oxygen species (ROS). In this condition, the FOXO transcription factor is translocated to the nucleus by, for example, c-Jun N-terminal kinase (JNK), resulting in the transcription of an alternative set of target genes of which the produced proteins protect the cell against the consequences of oxidative stress [33,41]. One of these target genes, superoxide dismutase 2 (*SOD2*/*MnSOD*), was successfully used to distinguish between the two states of FOXO transcriptional activity [67]. Oxidative stress was defined as high FOXO activity in combination with elevated *SOD2* mRNA level (defined as >2 standard deviations over the level in healthy non-dividing cells). In such a cellular stress condition, our PI3K pathway test cannot formally infer PI3K pathway activity. However, the PI3K pathway is most likely active and probably not the only active signaling pathway contributing to cell division, e.g., other growth factor pathways such as the MAPK and JAK-STAT pathways can also be activated [59,69].

Comparison of the here discussed PI3K pathway test with other approaches to develop a signature for pathway activity revealed important differences [64,67]. Multiple data-driven machine learning methods have been employed by various groups to discover growth factor pathway signatures, frequently using large “omics” datasets [70,71,72]. Resulting signatures consist of features (e.g., genes, mutations, proteins, etc.) associated with, but not causally related to, activity of a pathway, e.g., the PI3K pathway. Due to lack of causal relations between signature features and functional pathway activity, such a signature only indirectly relates to pathway activity. In addition, data driven approaches are unavoidably associated with a degree of overfitting which hampers performance on sample types unrelated to those used in the training set [73] (also discussed in [64]). 

The envisioned clinical use of the PI3K pathway test lies in predicting and monitoring the response to therapy, especially PI3K pathway inhibition, across cancer types. In the case of an active pathway, (targeted) mutation or copy number analysis (e.g., *PIK3CA*, *PTEN*, RTK such as *HER2*) may reveal a genomic cause for the identified activity of the PI3K pathway, providing the complementary information needed to choose the type of PI3K pathway inhibitor drug. Since PI3K, MAPK, and JAK-STAT pathways may co-activate each other depending on cell type and cellular condition, use of targeted MAPK and JAK-STAT pathway inhibitors may also result in a decrease in PI3K pathway activity score, reflecting pathway crosstalk [67]. 

PI3K pathway analysis is also expected to be of value to assess the functional activity status of a variety of immune cell types (e.g., lymphocyte subpopulations) in the anti-cancer immune response, which may be of help to predict response to immunotherapy [64,67]. 

It may also be useful in combination with the measurement of other signaling pathway activities to improve the characterization of tumor pathophysiology, to identify co-existing pathway activities and to provide information on their oncogenic versus tumor suppressive mode of action, as well as a potential role in therapy resistance [64,67]. As discussed above, the well-known tumor suppressive activity of the TGFβ signaling pathway can be either lost in the presence of an active PI3K pathway or converted to a tumor promoting pathway, which may confer therapy resistance [45,67,74,75]. Indeed, in advanced prostate cancer we frequently identified a combination of an active PI3K pathway with the loss of TGFβ pathway activity [67]. 

PI3K pathway activity has also been linked to chemoradiation resistance [59,76]. Many chemotherapeutic drugs, as well as radiation therapy, partly exert their cytotoxic effect through cellular generation of ROS, leading to DNA damage and apoptosis of the cell [59,77]. If an active PI3K pathway in rapidly dividing cancer cells generates oxidative stress, the FOXO-induced expression of enzymes like SOD2 is thought to provide protection against the DNA damaging consequences of oxidative stress, thereby preventing apoptosis [59]. By this mechanism, an active PI3K pathway may confer resistance to chemotherapy and radiation, which can be identified by the measured combination of high FOXO activity and high *SOD2* mRNA [67]. In the case of radiation therapy, the role of the PI3K pathway, in repair of double stranded DNA breaks, provides an additional resistance mechanism [76,78]. Taken together, we believe that measuring PI3K-FOXO activity in cancer samples will have clinical utility with respect to the prediction of therapy response and resistance. 

Finally, it was recently reported that many cell lines differ in their transcriptome, compromising the reproducibility of drug response experiments [79]. In life science research and drug development, measuring PI3K pathway activity, preferably in combination with measuring the activity of other signaling pathways, may be of value to quantitatively characterize and compare cell lines, and to compare cell (or xenograft mice) experiments to quantify and improve the reproducibility of experimental results [67]. 

## 5. Conclusions

The PI3K growth factor signaling pathway plays an important role in progression and treatment resistance of many types of cancer. Multiple drugs targeting the pathway are available and being developed. However, pathway activity varies on an individual patient basis and cannot be reliably assessed by currently available tests, including DNA mutation analysis. As a consequence, there is a need for improved tests to measure pathway activity. The here discussed mRNA-based PI3K pathway assay measures functional activity of the pathway in a quantitative manner. Upon further clinical validation it is expected to contribute to improved prediction of response to cancer treatments, including PI3K pathway targeting drugs, as well as be of use in drug development.

## Figures and Tables

**Figure 1 cancers-11-00293-f001:**
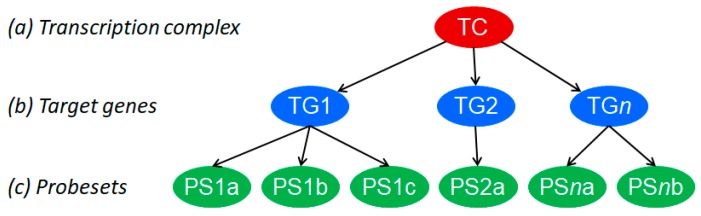
Knowledge-based Bayesian computational pathway model. The Bayesian network structure, used as a basis for our modeling approach and shown as a simplified model of the transcriptional program of a cellular signal transduction pathway, consists of three types of nodes: transcription factor, target gene, and microarray probe sets corresponding to the target gene. Used with permission from Reference [65].

**Figure 2 cancers-11-00293-f002:**
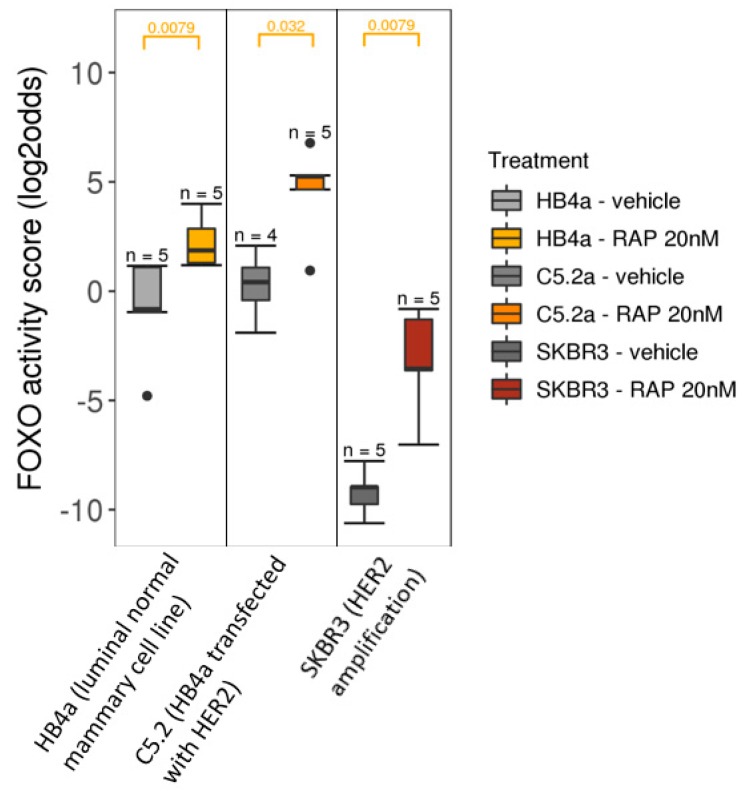
PI3K pathway activity analysis on Affymetrix U133Plus2.0 data of public GEO dataset GSE26599, containing samples of breast cancer cell lines treated with Rapamycin (mTOR inhibitor), indicated in the legend as RAP. Left panel: HB4a normal mammary epithelial cell line; middle panel: *HER2*-transfected HB4a cell line; right panel: *HER2*-amplified SKBR cell line. The FOXO activity score is inversely related to PI3K pathway activity. Wilcoxon signed-rank statistical test, p-value indicated at the top of the graph (orange).

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
