# Peer review of "Quantitative Measurement of Functional Activity of the PI3K Signaling Pathway in Cancer"

_cancers, 2019, doi:10.3390/cancers11030293_

Round 1
Reviewer 1 Report
The perspective focuses on the author’s own recent work describing the development of a predictive PI3K pathway activity computational assay. The work covers the background on the role of the PI3K pathway in cancer thoroughly and places the context of measuring FOXO transactivation as a surrogate for PI3K activity. The perspective would have been more interesting and useful if other examples of using similar computational methods for other pathways were covered (e.g. Way et al., https://doi.org/10.1016/j.celrep.2018.03.046; Vaske et al., https://doi.org/10.1093/bioinformatics/btq182).
Figure 1 is not linked to anything mentioned in the text; this needs to be addressed to place Figure 1 in context.
On page 4, line 155, I’m not clear what ‘ground truth’ means; I’ve never heard this before. I guess that the author means ‘ground state’ or ‘basal state’.
Page 4, Line 167, should ‘Oxidative cell..’ be “Oxidative stress..”?
There are many grammatical errors throughout the manuscript:
For example, in Abstract line 10 ‘causing cell division’ should be ‘driving’ or ‘promoting’.
Introduction, line 29 ‘evolutionary’ should be ‘evolutionarily’.
Line 55, delete ‘Simplified’.
Line 94, ‘frequent’ should be ‘common’.
Line 109, delete ‘make dominant’.
Line 132, delete ‘relatively’.
Line 133, ‘deranging’ should be ‘disrupting’.
Lone 182, ‘for the PI3K activity’, should this be ‘for changes in PI3K activity’?
Author Response
Referee 1:
I would like to thank the referee for taking the time to carefully review the manuscript, the comments were very useful to improve the quality. I believe I have been able to address them sufficiently.
Below you find my response, added below each comment.
The perspective focuses on the author’s own recent work describing the development of a predictive PI3K pathway activity computational assay. The work covers the background on the role of the PI3K pathway in cancer thoroughly and places the context of measuring FOXO transactivation as a surrogate for PI3K activity. The perspective would have been more interesting and useful if other examples of using similar computational methods for other pathways were covered (e.g. Way et al., https://doi.org/10.1016/j.celrep.2018.03.046; Vaske et al., https://doi.org/10.1093/bioinformatics/btq182).
Reply: I have added a paragraph on other signature-based approaches to measure PI3K pathway activity. To the best of our knowledge there is no diagnostic test available which measures functional PI3K pathway activity to compare with. Added text + references: “Comparison of the here discussed PI3K pathway test with other approaches to develop a signature for pathway activity revealed important differences [64],[67]. Multiple data-driven machine learning methods have been employed by various groups to discover growth factor pathway signatures, frequently using large “omics” datasets [70],[71],[72]. Resulting signatures consist of features (e.g., genes, mutations, proteins, etc.) associated with, but not causally related to, activity of a pathway, e.g., the PI3K pathway. Due to lack of causal relations between signature features and functional pathway activity, such a signature only indirectly relates to pathway activity. In addition, data driven approaches are unavoidably associated with a degree of overfitting which hampers performance on sample types unrelated to those used in the training set [73] (also discussed in [64]). “
Figure 1 is not linked to anything mentioned in the text; this needs to be addressed to place Figure 1 in context.
Reply: This is done.
On page 4, line 155, I’m not clear what ‘ground truth’ means; I’ve never heard this before. I guess that the author means ‘ground state’ or ‘basal state’.
Reply: We use this to indicate a known state (a confirmed truth, ground truth) of pathway activity. I have changed it into “the state of FOXO activity was known”.
Page 4, Line 167, should ‘Oxidative cell..’ be “Oxidative stress..”?
Reply: This has been corrected.
There are many grammatical errors throughout the manuscript:
Reply: The manuscript has been checked for grammatical errors by MPDI English editing service.
For example, in Abstract line 10 ‘causing cell division’ should be ‘driving’ or ‘promoting’.
Reply: This has been changed.
Introduction, line 29 ‘evolutionary’ should be ‘evolutionarily’.
Reply: Done.
Line 55, delete ‘Simplified’.
Reply: Done.
Line 94, ‘frequent’ should be ‘common’.
Reply: Done.
Line 109, delete ‘make dominant’.
Reply: Done.
Line 132, delete ‘relatively’.
Reply: Done.
Line 133, ‘deranging’ should be ‘disrupting’.
Reply: Done.
Lone 182, ‘for the PI3K activity’, should this be ‘for changes in PI3K activity’?
Reply: I have tried to clarify the meaning of this sentence as follows: “Envisioned clinical use of the PI3K pathway test lies in predicting and monitoring response to therapy, especially PI3K pathway inhibition, across cancer types. In case of an active pathway, (targeted) mutation or copy number analysis (e.g, PIK3CA, PTEN, RTK like HER2,) may reveal a genomic cause for the identified activity of the PI3K pathway”.
Reviewer 2 Report
This perspective-type review is based on previous publications from the author reporting the development of an elegant approach to measure PI3K pathway activity. Although this would be of general interest, the review is not sufficiently well focused and organised as it stands for the purpose of a perspective. The amount of information on general signalling pathways should be reduced to allow a better focus on the PI3K pathway. In contrast, sections on FOXO-mediated transcription as well as other existing approaches for the sake of comparison should be expanded.
Minor comments:
Line 39. Something is missing in this sentence
Line 62. the human gene name PI3KCA should be in italic. Amend all other gene names in the text.
Line 62. The gene names for p110b, d and g should also be stated.
Line 63: phosphatidylinositol trisphosphate should be written as phosphatidylinositol (3,4,5) triphosphate and abbreviated as PtdIns(3,4,5)P3 (with P as italic and 3 underlined).
Line 73. The full name of FOXO should appear earlier on line 57.
The text should be thoroughly checked for grammar and minor wording details.
Author Response
Referee 2:
I would like to thank the referee for taking the time to carefully review the manuscript, the comments were very useful to improve the quality. I believe I have been able to address them sufficiently.
Below you find my response, added below each comment.
This perspective-type review is based on previous publications from the author reporting the development of an elegant approach to measure PI3K pathway activity. Although this would be of general interest, the review is not sufficiently well focused and organised as it stands for the purpose of a perspective. The amount of information on general signalling pathways should be reduced to allow a better focus on the PI3K pathway. In contrast, sections on FOXO-mediated transcription as well as other existing approaches for the sake of comparison should be expanded.
Reply: Since activity of the PI3K pathway is involved in regulation/activation of many of the other signaling pathways (as explained), it is of importance to provide some basic information on the other signal transduction pathways. I modified the following sentence to make this clear and added 2 references (other are added later in the manuscript when these interactions are described in more detail): “From a clinical perspective, the PI3K/AKT/mTOR signal transduction pathway is certainly one of the most important signaling pathways, as it regulates key processes such as cell division, migration, metabolism, and DNA repair in a large variety of cell types, to an important extent also by recruiting and influencing activity of one or more other signaling pathways [30],[31],[32],[33]”
Later in the manuscript I have modified a sentence to explain the interaction between FOXO and SMAD: “As an example, the TGFβ pathway exerts an anti-proliferative effect in the absence of an active PI3K pathway, mediated by transcriptional activity of the TGFβ-associated SMAD transcription factor together with FOXO. Upon activation of the PI3K pathway, this tumor suppressive effect of TGFβ can be lost due to cytoplasmic translocation of FOXO, or even switched towards a tumor-promoting effect in the presence of an active MAPK-AP1 pathway [43],[44],[45].”
I have added an additional reference for regulation of functional FOXO activity (Hornsveld et al, ref. 33), and adapted this sentence: “Oxidative stress can be caused by an active PI3K pathway (potentially in combination with other growth factor pathways) causing rapid cell division and metabolic changes, resulting in high levels of reactive oxygen species (ROS). In this condition the FOXO transcription factor is translocated to the nucleus by for example c-Jun N-terminal kinase (JNK), resulting in transcription of an alternative set of target genes of which the produced proteins protect the cell against the consequences of oxidative stress [33],[41]..”
The introductory part on FOXO has been expanded (including references): “The PI3K pathway is linked via the activation of AKT kinase to FOXO transcription factors. FOXO family members are ubiquitously expressed transcription factors [31],[33],[38],[39],[40]. They transcribe their target genes by recognizing and binding to two different promoter enhancer sites, the Daf-16 family member-Binding Element (DBE, 5′-GTAAA(T/C)AA-3′) and the Insulin-Responsive Element (IRE, 5′-(C/A)(A/C)AAA(C/T)AA-3′) [41]. In healthy non-dividing tissue, FOXO’s are located in the nucleus and when active they transcribe target genes that are necessary to maintain a non-dividing differentiated cellular phenotype, e.g., multiple cell cycle kinase inhibitors [33].”.
I think that more detailed description of FOXO regulation is outside the scope of this perspective.
I have added a paragraph on other signature-based approaches to measure PI3K pathway activity. To the best of our knowledge there is no diagnostic test available which measures functional PI3K pathway activity to compare with. Added text + references: “Comparison of the here discussed PI3K pathway test with other approaches to develop a signature for pathway activity revealed important differences [64],[67]. Multiple data-driven machine learning methods have been employed by various groups to discover growth factor pathway signatures, frequently using large “omics” datasets [70],[71],[72]. Resulting signatures consist of features (e.g., genes, mutations, proteins, etc.) associated with, but not causally related to, activity of a pathway, e.g., the PI3K pathway. Due to lack of causal relations between signature features and functional pathway activity, such a signature only indirectly relates to pathway activity. In addition, data driven approaches are unavoidably associated with a degree of overfitting which hampers performance on sample types unrelated to those used in the training set [73] (also discussed in [64]). “
Minor comments:
Line 39. Something is missing in this sentence
Reply: I have increased the readability of this sentence as follows: Naturally, in physiological cellular processes, signaling pathway activity as well as crosstalk between different pathways, is tightly controlled [15],[16],[17],[18].
Line 62. the human gene name PI3KCA should be in italic. Amend all other gene names in the text.
Reply: Done.
Line 62. The gene names for p110b, d and g should also be stated.
Reply: PIK3CB, PIK3CD, PIK3CG gene names were added.
Line 63: phosphatidylinositol trisphosphate should be written as phosphatidylinositol (3,4,5) triphosphate and abbreviated as PtdIns(3,4,5)P3 (with P as italic and 3 underlined).
Reply: Done.
Line 73. The full name of FOXO should appear earlier on line 57.
Reply: Done.
The text should be thoroughly checked for grammar and minor wording details.
Reply: The manuscript was checked for grammatical errors by MPDI English editing service.
Round 2
Reviewer 2 Report
I am overall satisfied with the answers of the author. The text has been improved and it now reads a lot better.